# Definite photon deflections of topological defects in metasurfaces and symmetry-breaking phase transitions with material loss

Chong Sheng[1], Hui Liu [ID][1], Huanyang Chen[2] & Shining Zhu[1]

Combination of topology and general relativity can lead to some profound and farsighted predictions. It is well known that symmetry breaking of the Higgs vacuum field in the early universe possibly induced topological defects in space-time, whose nontrivial effects can provide some clues about the universe's origin. Here, by using an artificial waveguide bounded with rotational metasurface, the nontrivial effects of a topological defect of spacetime are experimentally emulated. The photon deflection in the topological waveguide has a robust definite angle that does not depend on the location and momentum of incident photons. This is remarkably different from the random optical scattering in trivial space. By including material loss such a topological effect can be well understood from the symmetry breaking of photonic modes. Our technique provides a platform to investigate topological gravity in optical systems. This method can also be extended to obtain many other novel topological photonic devices.

[1] National Laboratory of Solid State Microstructures and School of Physics, Collaborative Innovation Center of Advanced Microstructures, Nanjing University, Nanjing 210093 Jiangsu, China. [2] Department of Electronic Science, Institute of Electromagnetics and Acoustics, Xiamen University, Xiamen 361005, China. Correspondence and requests for materials should be addressed to H.L. (email: liuhui@nju.edu.cn)

Topology plays a very important role in the frontiers of modern physics. It is well known that topological properties have been widely reported in condensed matter[1–4] and classical waves[5–8]. In addition, topology is also employed in general relativity and modern cosmology[9]. There will be increased interest in the topological effects of cosmic space-times due to the recent breakthrough of gravitational wave detection. In particular, theorists predicted that some topological defects may have formed during a symmetry breaking of the Higgs vacuum field in the early universe when the topology of the vacuum manifold associated with this symmetry breaking was not simply connected. Examples are monopoles, cosmic strings, and domain wall[10], which respectively are 0-dimensional, 1-dimensional, and 2-dimensional topological defects of spacetime. The experimental observation of these topological defects in astrophysics would revolutionize the vision of the cosmos. Furthermore, various schemes, such as gravitational waves and specific imprints in cosmic microwave background (CMB) radiation generated by cosmic strings, might be proposed for astronomical observation. However, to the best of our knowledge, thus far there is no good way to observe these topological defects in astrophysics. Fortunately, analog models from various systems in the laboratory environment have been motivated by the possibility of investigating phenomena not readily accessible in their cosmological counterparts. Examples are Hawking–Unruh radiation emulated from a Fermi-degenerate liquid[11], a superconductor[12], optical fiber[13], nonlinear crystal[14], ion ring[15], and Bose–Einstein condensates[16].

On the other hand, transformation optics based on metamaterials that manipulate permittivity and permeability profiles can be extensively investigated to design many artificial materials with novel optical applications[17–29]. The most important application of transformation optics is invisibility cloaks, in which light is regarded as linear parallel geodesic rays in deformed spaces. In Einstein's general relativity theory, the spacetime curvature is determined by the energy and momentum. Recently, through mapping the metric of spacetime to electromagnetic medium constitutive parameters with local curvature, it has become possible to mimic general relativity phenomena. Examples from general relativity are black holes[30–35], Einstein ring[36], cosmic string[37–39], Minkowski spacetimes[40], electromagnetic wormholes[41], De-Sitter Space[42], cosmological inflation, and redshift[43]. The underlying principle is the form invariance of Maxwell's equations between the complex inhomogeneous media and the background of an arbitrary spacetime metric. Recently, some other optical structures, such as curved waveguides[44,45], nonlocal media[46,47], and optical lattices[48,49], have also been used to simulate the gravitational effects.

In addition to the 3-dimensional metamaterials, metasurface (2-dimensional metamaterials), with easy fabrication and low propagation loss, represent a new paradigm to manipulate electromagnetic waves. Examples are generalized refraction[50,51], optical spin-Hall effects[52,53], nonlinear response[54,55], flat metalenses[56], and information coding[57]. In particular, different kinds of metasurface have been used to control the propagation of surface waves[58–66].

In this work, we experimentally demonstrate that a cosmic string, a 1-dimensional topological defect of space, can be emulated in an artificial waveguide, and its nontrivial topological effects can be directly observed. Given the corresponding relation between electromagnetic parameters and the metric of spacetime in general relativity, the electromagnetic parameters required by a cosmic string are equivalent to those of a coiled uniaxial crystal with a rotation axis being the defect center. Such an effective medium with a singularity was practically realized in an artificial waveguide bounded by a rotational metasurface, and its optical properties were investigated by exciting the waveguide modes. In our experiments, we can observe the photon deflection process in the synthetic topological space with a robust deflection angle that does not depend on the locations and momentum of incident photons. This is remarkably different from ordinary optical scattering in trivial space. Such a topological property can be regarded as the consequence of the symmetry breaking of photonic modes after including material loss. By tuning the structure parameters, we can obtain a symmetry breaking phase transition from trivial space to nontrivial topological space.

## Results

**Theoretical design of synthetic topological space.** To emulate the gravitational lensing of topological defects in cosmology, we start with a spacetime metric for a static cylindrically symmetric cosmic string (see Fig. 1a) in the low-mass limit with straight and infinite long length given as[67]:

$$ds^2 = dt^2 - dr^2 - \alpha^2 r^2 d\varphi^2 - dz^2 \tag{1}$$

where $\alpha = 1 - 4G\mu$, $G$ is the gravitational constant, $\mu$ is the linear mass density of the string along the rotation axis, and natural units have been adopted. The corresponding Riemann curvature is given by $R_{12}^{12} = 2\pi(1-\alpha)\delta^{(2)}(r)/\alpha$, where $\delta^{(2)}(r)$ is the 2-dimensional Dirac delta function in the plane. Therefore, the string is locally flat except a conical singularity at the origin. According to Riemann curvature, if the mass density of the string $\mu > 0$, it carries positive curvature at the origin; and, it carries negative curvature with a negative mass density ($\mu < 0$). It is well known that the motion of the photon follows the null geodesic of space-time, thus the trajectory of light moves toward (away from) the origin of the string with constant negative (positive) deflection angle, as shown in Fig. 1b–d. According to the geodesic equation: $\ddot{x}^\lambda + \Gamma^\lambda_{uv}\dot{x}^u\dot{x}^v = 0$, where $x$ represents the spatial coordinates and the derivatives are taken over an arbitrary affine parameter. The metric of the string has nonzero Christoffel symbols $\Gamma^r_{\varphi\varphi} = -\alpha^2 r$, $\Gamma^\varphi_{r\varphi} = \Gamma^\varphi_{\varphi r} = 1/r$, and the light trajectory after some calculation can be described by $r^2\alpha^2\dot{\varphi}^2 = C\dot{r}^2$, where $C = b^2/(r^2 - b^2)$, and $b$ is the impact parameter. Therefore, we can analytically calculate the trajectory under different impact parameters. Furthermore, based on the Gauss–Bonnet theorem[68], the light deflection angle is $\Delta\theta = \pi(1-\alpha)/\alpha$ and is independent of the impact parameter.

The nontrivial topological space of a cosmic string can be simulated in an effective medium, according to the correspondence between the effective electromagnetic parameters and the spacetime metric based on transformation optics[19]: $\varepsilon^{ij} = \mu^{ij} = \sqrt{-g}g^{ij}/(\sqrt{\zeta}g_{00})$, where $\varepsilon^{ij}$ and $\mu^{ij}$ are artificial material parameters, $g^{ij}$ is the metric of the cosmic string based on equation (1), $\zeta$ is the determinant of a spatial metric to transform one set of spatial coordinates to another. After calculations, the material parameter tensors required by the cosmic string have their principal values along $r$, $\varphi$, and $z$ directions given as[69]

$$\varepsilon_r = \mu_r = \alpha, \varepsilon_\varphi = \mu_\varphi = 1/\alpha, \varepsilon_z = \mu_z = \alpha \tag{2}$$

which can also be obtained by performing a linear transformation along the $\varphi$ direction from a Minkowski space $ds'^2 = dt'^2 - dr'^2 - r'^2 d\varphi'^2 - dz'^2$, with $r' = r$, $\varphi' = \alpha\varphi$, $z' = z$. Here, considering different polarized light, we can define two orthogonal polarized waves: transverse electric (TE) wave with fields $(E_\varphi, E_r, H_z)$ and transverse magnetic (TM) wave with fields $(H_\varphi, H_r, E_z)$. For TE wave, we can take the refractive indices[69] $\left(n_\varphi^2 = \varepsilon_r\mu_z = \alpha^2, n_r^2 = \varepsilon_\varphi\mu_z = 1\right)$. For TM wave, we can take the refractive indices[70] $\left(n_\varphi^2 = \mu_r\varepsilon_z = \alpha^2, n_r^2 = \mu_\varphi\varepsilon_z = 1\right)$. Thus,

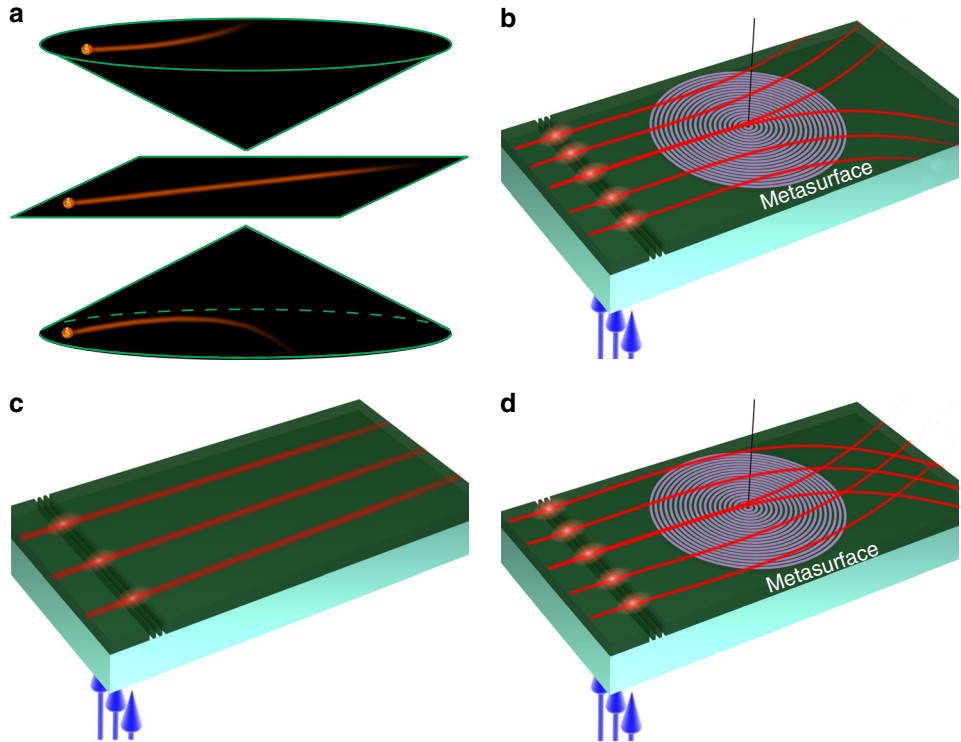

**Fig. 1** Mimicking definite photon deflections in topological spaces using artificial waveguides. **a** Photon deflection in the space of negative topological defect (photons pushed away from the center), positive topological defect (photons attracted to the center), and trivial space (photons pass along straight line). Definite photon deflection in artificial waveguide with negative topological defect (**b**), positive topological defect (**d**), and no topological defect (**c**). In (**b, c, d**), the blue arrows represent the incident laser beam from the bottom and the red lines represent the light rays inside the waveguides

for both the polarizations, the corresponding refractive indices of the cosmic string are equivalent to a uniaxial crystal with a rotating axis $n = \begin{pmatrix} n_\varphi & 0 \\ 0 & n_r \end{pmatrix}$.

It is usually very difficult and impractical to construct an effective medium given equation (2) based on traditional 3-dimensional metamaterials in the optical region. In this work, we find a way to construct such a medium by combining a slab waveguide with a rotational metasurface (see Fig. 2a, b). The metasurface which consists of a subwavelength grating, can be considered to weakly disturb the waveguide modes. The effective parameter of such a waveguide is described as a tensor:

$$n^2 = R^{-1}(\varphi) \begin{bmatrix} n_e^2 & 0 \\ 0 & n_o^2 \end{bmatrix} R(\varphi), \text{ where } R(\varphi) = \begin{bmatrix} \cos(\varphi) & \sin(\varphi) \\ -\sin(\varphi) & \cos(\varphi) \end{bmatrix}$$

is the rotating operator, $n_o$ and $n_e$ are the waveguide effective indices along the radial and azimuthal direction, respectively. As a whole region, such a waveguide is non-uniform whose effective index is angular dependent (see Fig. 2c). But in a local region (see Fig. 2d), the waveguide can be seen as a uniform anisotropic medium with a local elliptical iso-frequency contour: $(k_r/n_o)^2 + (k_\varphi/n_e)^2 = 1$, where $k_r$ and $k_\varphi$ are wave vectors along the radial and azimuthal direction (see Supplementary Note 2). The local waveguide mode index has two principle values $n_e$ and $n_o$, whose effective index distribution is equivalent to that of a uniaxial crystal. By considering reduced refractive index, we see that the effective parameters in a structured waveguide using metasurface as the circular grating correspond well to that required by a cosmic string. And the ellipticity $\eta = n_e/n_o$ in iso-frequency contour plays the same role as the parameter $\alpha$ in the metric of the string. If $\eta < 1$, the string has positive mass density and carries positive curvature at the origin; and the photon undergoes an attractive force and moves toward to the origin of the string

(Fig. 1d). When $\eta > 1$, the string has negative mass density and carries negative curvature at the origin, and the photon undergoes a repulsive force and moves away from the origin of the string (Fig. 1b). Furthermore, for the case of the isotropic waveguide, the ellipticity becomes $\eta = 1$, which corresponds to the absence of defect; the curvature of spacetime is zero; thus the photon undergoes no force, and light propagates in a straight line (Fig. 1c). The very remarkable property of such a waveguide is that the light deflection angle is only determined by the parameter $\eta$, which originates from the intrinsic topological properties of the cosmic string, which does not depend on the location and momentum of incident photons.

**Experiments of photonic deflection in topological space.** In our experiments, the rotational metasurface with a set of periodic subwavelength circular gratings (as shown in Fig. 2b) was fabricated by focused ion beam (FEI Strata FIB 201, 30 keV, 7.7 pA) milling on a 200-nm-thickness silver film, which has been initially deposited on a silica substrate ($SiO_2$). Then a polymethylmethacrylate (PMMA) resist mixed with oil-soluble PbS quantum dots was deposited on the silver film using a spin-coating process (as shown in Fig. 2a). Figure 2a shows the $SiO_2$/silver/metasurface/PMMA/air multilayer structure. Figure 2c is the schematic plot of the structure, which indicates locally periodic gratings along the $r$ direction as shown in Fig. 2d. In one fabricated sample, the structural parameters (see Supplementary Figure 1 (d)) are chosen as: the grating period $p = 120$ nm, the silver filling ratio $f = 0.58$, and thickness $t = 200$ nm, the depth $h = 60$ nm, PMMA thickness $d = 300$ nm; and the calculated effective parameters $n_e = 1.30$, $n_o = 1.22$, thus the ellipticity $\eta \approx 1.065 > 1$. This corresponds to a cosmic string with negative mass density and negative curvature at the origin; thus the motion of

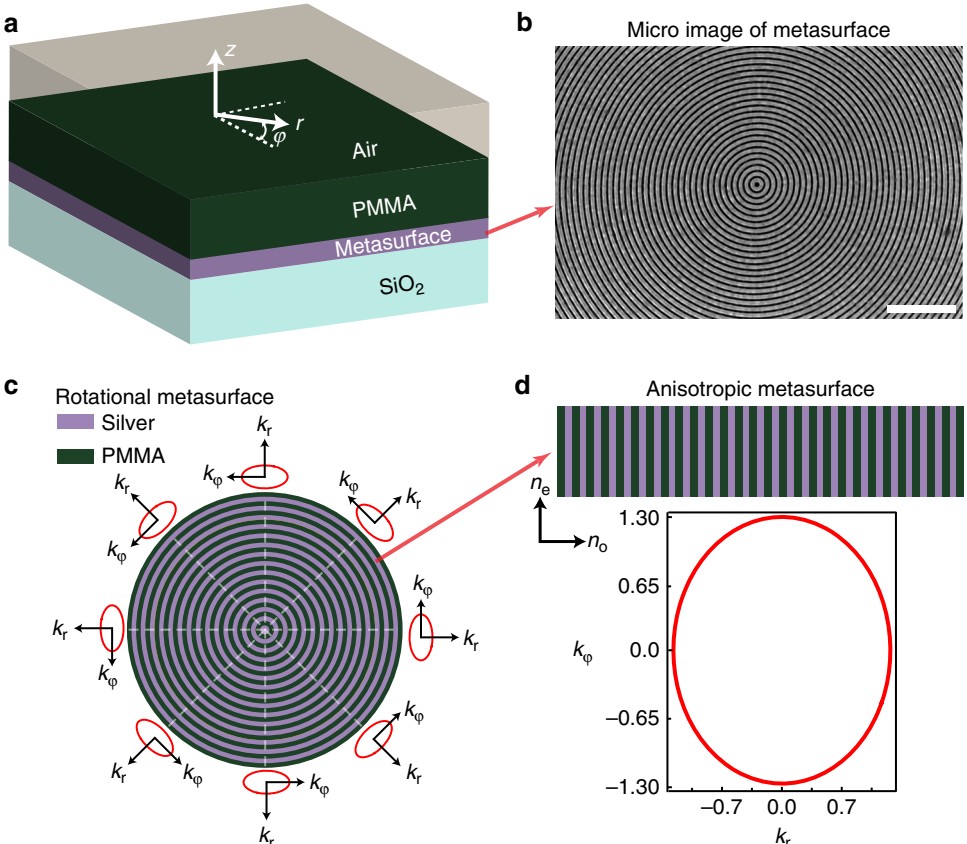

**Fig. 2** Theoretical design of artificial waveguides. **a** Schematic of a multilayer artificial waveguide. A cylindrical coordinate system ($z,r,\varphi$) is defined. **b** A micro-image of a rotational metasurface fabricated with a focused ion beam. The length of white scale bar is 1 μm. **c** The rotational metasurface with local anisotropic indices which are denoted by red curves. **d** The local elliptical iso-frequency contour of the artificial waveguide

light undergoes a repulsive force and moves away from the origin of the string, as shown in Fig. 3a–f. According to the above theory, the light deflection in the synthetic nontrivial topological space is independent of the impact parameter with a definite angle $\theta_1 = (\eta - 1)*\pi/\eta = 11.0°$. This behaves drastically different from optical scatterers in trivial space usually that show broad random scattering angles. Such a nontrivial property can be directly observed in our experiment.

To study light propagation in an artificial waveguide, 785 nm light from a continuous-wave (c.w.) laser was coupled into the waveguide through a set of gratings with 650 nm period (see Fig. 1b); and the TE modes are chosen when the polarization of laser is parallel to coupling grating. As the coupled light propagates within the waveguide, it excites the quantum dots, which then re-emit fluorescent light at 1050 nm. The fluorescence emission will reveal the propagation dynamics and is used to analyze the ray trajectory. In the experimental process, the impact parameter of the incident photon is tuned by moving the location of the laser spot on the grating. The optical deflection in the waveguide was observed through a fluorescence imaging technique. The results under three different impact parameters are illustrated in Fig. 3a–c. And the measured deflection angles of three cases have almost the same value $\theta_1 = 11.0°$. Furthermore, to validate our experimental finding, full-wave simulations are performed using the finite element software COMSOL Multi-physics. The calculated deflection profiles are given in Fig. 3d–f. Compared with calculation results, the experiment results are not very clear which is mainly caused by the experiment techniques used in our work. Firstly, the quantum dots are not distributed

uniformly inside the PMMA layer. Some quantum dots aggregate together and make the fluorescent intensity emitted at different locations non-uniform. Secondly, the size of the structure is very small and close to the ultimate machining precision of FIB. Some structure defects are unavoidably induced in the fabrication process. These defects will cause random scattering of light inside the waveguide. Although the quality of the experiment picture is not ideal due to the above experiment processes, the deflection angles obtained in Fig. 3a–c can be still determined and compared with theoretical results in Fig. 3d–f.

In addition to the negative curved curvature at the origin of the cosmic string with a negative mass density, the string with a positive mass density can also be experimentally demonstrated in such a system. In another experiment, the structural parameters are chosen as $f = 0.58$, $p = 120$ nm, $h = 45$ nm, $t = 200$ nm and PMMA thickness $d = 0$ (see Fig. 2a, b). For such a structure, we only have the first order TM modes, which are usually known as surface plasmon modes. Based on waveguide theory, the ellipticity is obtained as $\eta = 0.92$ ($n_e = 1.03$, $n_o = 1.12$). It corresponds to the cosmic string with positive mass density, and the light propagates toward the origin of the string with a deflection angle $\theta_1' = -15.7°$ under different impact parameters. Based on the same optical technique used above, we experimentally investigated the light deflection under different impact parameters. We still use 785 nm laser as the input light. The measured results are shown in Fig. 3g–i, in which the theoretical calculated geodesic lines are given as green dashed lines for comparison. The simulated results are shown in Fig. 3j–l. Although the experiment results are not very clear due to experimental technique problems,

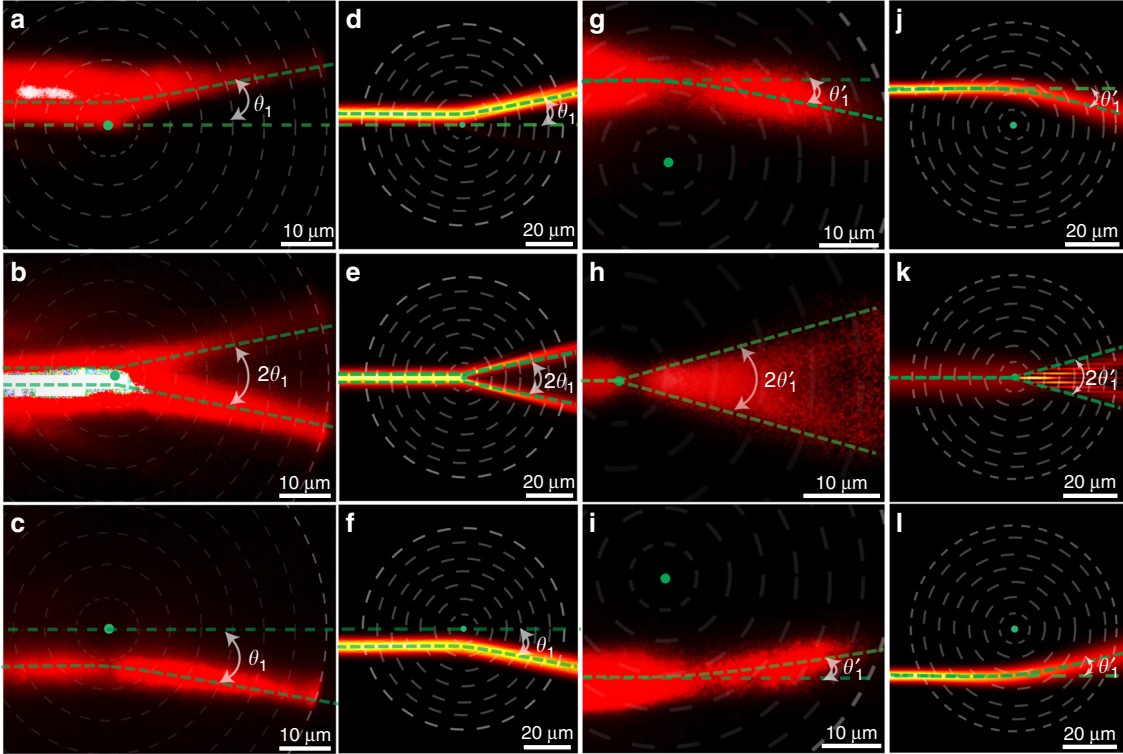

**Fig. 3** Experiments and simulations of definite photon deflections in artificial waveguides. **a–c** Photon deflection observed in the experiment under different impact parameters ($b = 4.84$, 0, $-7.14$ μm) with negative mass density $\eta \approx 1.065$. **d–f** The corresponding simulations using COMSOL Multiphysics. All three cases have the same deflection angle $\theta_1 = 11.0°$. **g–i** Photon deflection observed in the experiment under different impact parameters ($b = 12.91$, 0, $-17.06$ μm) with positive mass density $\eta \approx 0.92$. **j–l** The corresponding calculated results using COMSOL Multiphysics. All three cases with positive mass density have the same deflection angle $\theta'_1 = -15.7°$. The green dashed line indicates the calculated deflection angle from theory, the dashed white circles represent metasurface zones, and the green dot is the location of defect. In order to magnify the detail of the optical field profile and easily determine the deflection angle, the size of experiment images is zoomed in and the brightness contrast is increased

it can be still seen that the deflection angles do not change under three different incident impact parameters, which agree with theoretical and simulation results.

**Symmetry breaking of photonic modes with material loss.** According to quantum field theory, a cosmic string originates from the topological phase transition induced by spontaneous symmetry breaking of the Higgs vacuum field in the early universe. In our system, such a symmetry breaking can be mimicked through the symmetric breaking of photonic modes after including material loss. In the above discussion, we can see that the non-degeneracy of TE and TM modes play a very important role in obtaining the topological space. Then, if we can find some way to transform the non-degenerate modes to degenerate modes in our system, it will be possible for us to mimic the birth of a cosmic string during the spontaneous symmetric breaking phase transition. Usually, the momentum of the TE mode $k_{TE}$ and TM mode $k_{TM}$ cannot equal each other, and there always is a momentum mismatch $\Delta k = |k_{TE} - k_{TM}|$, which is related to a coherence length $l_c = 1/\Delta k$. For a finite value of $l_c$, we can always differentiate the TE mode from the TM mode because the light can propagate a very long distance ($>l_c$) without loss. However, if we include material loss, the propagation distance will have a finite value. Here, because $l_{TE} \neq l_{TM}$, we define the propagation distance $l_d$ as the longer length of $l_{TE}$ and $l_{TM}$ for convenience. Now, if the propagation distance is smaller than the coherence length $l_d < l_c$, we cannot differentiate TE from TM mode and the waveguide modes are in degenerate states. This result can also be understood from Heisenberg's uncertainty principle $\Delta x \cdot \Delta p \geq \hbar/2$. With material loss, the variation range of the

location of a photon inside the waveguide is determined by the propagation distance $\Delta x \sim l_d/2$. The allowed minimum variation of momentum is $\Delta p_{min} \sim \hbar/l_d$. If $l_d > l_c$, the momentum mismatch $\hbar \Delta k = \hbar/l_c > \Delta p_{min}$, and the two modes can be distinguished. However, if $l_d < l_c$, we have $\hbar \Delta k < \Delta p_{min}$, and the TE and TM modes cannot be distinguished and merge to one degenerate mode. Here, the critical case $l_d = l_c$ can be regarded as the symmetric breaking condition. In addition, the degenerate photonic modes of $l_d < l_c$ can be seen as the symmetric phase ($k_{TE} = k_{TM}$) and the non-degenerate states of $l_d > l_c$ can be seen as the symmetry breaking phase ($k_{TE} \neq k_{TM}$).

In our system, we can investigate the symmetry breaking process precisely through tuning the competition between $l_d$ and $l_c$. In order to obtain such a process, we will simultaneously change the material loss and waveguide thickness. Based on the waveguide theory, the wave vectors of the TE mode $k_{TE}$ and TM mode $k_{TM}$ depend on the PMMA thickness (see Supplementary Figure 1). Then we can calculate the thickness dependence of the coherence length $l_c = 1/|k_{TE} - k_{TM}|$ shown as the yellow solid line in Fig. 4a. Because the wave vector changes very fast for small $d$, we use $\delta = 1/d$ instead of $d$ in the discussion. In order to include loss, we define the refractive index of PMMA as $n = 1.49 + i\gamma$, where $\gamma$ represents the loss of PMMA. With the loss changing, we calculate the propagation distances of the TE and TM modes as illustrated in Fig. 4a. It can be seen that the yellow curve divides the figure into two regions. In the left region $l_d < l_c$, and this corresponds to the symmetric phase. While in the right region $l_d > l_c$, and this corresponds to the symmetric breaking modes. On the boundary (the yellow curve), the intersection point $\delta_c(l_c = l_d)$ can be seen as the critical phase transition point between the two

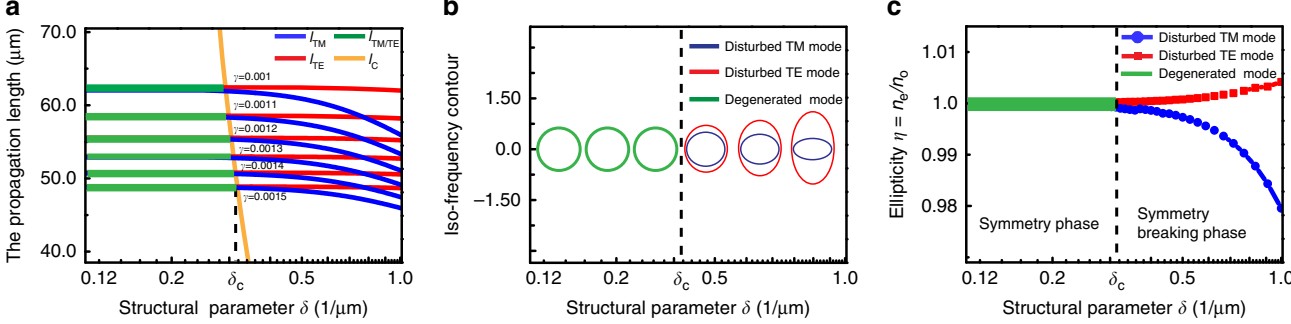

**Fig. 4** Symmetry breaking phase transition under different material loss. **a** The dependence of the propagation length of TE mode $l_{TE}$, TM mode $l_{TM}$, and coherence length $l_c$ on the structural parameter $\delta = 1/d$ ($d$ is the thickness of the PMMA layer) with different loss parameters $\gamma$. The yellow line splits the region into two parts: symmetric phase (on the left side) and symmetry breaking phase (on the right side). The dependence of iso-frequency (**b**) and ellipticity (**c**) of photonic modes on $\delta$ with phase transition point $\delta_c = 0.335\ \mu m^{-1}$. Here, material loss $\gamma = 1.5 \times 10^{-3}$

phases. In the early universe, when the temperature is cooling down, the cosmic string is created through symmetry breaking of the Higgs vacuum field. Here, we can see the parameter $\delta_c$ plays the role of critical temperature in the phase transition. In addition, this critical point $\delta_c$ can be tuned through changing the loss $\gamma$. In Fig. 4a, we give the $l_d$ with different loss $\gamma$. It can be seen that $l_d$ is reduced with increasing $\gamma$. As a result, the phase transition point $\delta_c$ is shifted (see Supplementary Figure 2 and Note 3). Here, the loss $\gamma$ plays the role as the self-energy of the Higgs field which determines the phase transition temperature. With the loss $\gamma = 0.0015$, the iso-frequency contour of the symmetric phase is a circle (see Supplementary Movie 1) with ellipticity $\eta = 1$ (see Fig. 4b). The iso-frequency contours of the symmetry breaking phase are two ellipses (see Supplementary Movie 1). The ellipticity of one curve is greater than unity ($\eta > 1$), which mimics the negative topological defect and the other is less than unity ($\eta < 1$), which mimics the positive topological defect (see Fig. 4b). If the loss disappears ($\gamma = 0$), the $l_d$ will become infinite and we will never obtain the transition point. Lastly, we can see that the material loss is the key in our system, which finally determines the symmetry breaking of spaces.

**Topological defects with different mass.** Furthermore, the cosmic string with various values of linear mass density can also be mimicked by steering the effective index profile of the waveguide through changing metasurface structures. In our process, we only tuned the depth $h$ of the metasurface (see Supplementary Figure 1 (b)) with other structural parameters unchanged. After the simulation calculation, it can be seen that the ellipticity of the effective parameters increases as the depth of the metasurface thickens. The larger ellipticity corresponds to the cosmic string with larger mass density at the origin, which leads to a larger deflection angle. As a result, the deflection angle will also increase with the thicker depth. In our experiment, we fabricated three samples with depths $h = 60, 52, 36$ nm. The measured results are provided in Fig. 5a–c, which are compared with numerical simulations in Fig. 5d–f. Although the experimental patterns are not very clear, we can still determine the deflection angles which agree with the theoretical and numerical results.

**Optical interference and accelerating beam in topological space.** Because the cosmic string has the novel property of a robust definite deflection angle that does not depend on the location and momentum of incident photons, we can design many definite optical devices which bend light without distorting the field pattern. As is well known, the optical interference is very sensitive to optical scattering, and the interference pattern can be

easily distorted by random scatterers in space. However, during the deflection process in the topological space of the cosmic string, the interference pattern between light beams is definitely bent without any obvious distortion. Figure 5g, h respectively shows the experimental results of optical interference in trivial space and topological space of the cosmic string. Figure 5k, l provides the corresponding simulations. It can be seen, in topological space, the interference pattern is uniformly bent with a definite angle $\theta_4 = 11.5°$ without obvious distortion.

Furthermore, the special property with the robust definite deflection angle can be also used to bend some other complex optical field, such as an accelerating electromagnetic (EM) wave packet. An example is the Airy beam, a paraxial beam that propagates along a parabolic trajectory while preserving its shape. These accelerating wave packets were followed by many applications such as manipulating microparticles, self-bending plasma channels, and accelerating electron beams. However, due to the realization condition required by the paraxial approximation, the accelerating angle is very small (see Fig. 5i, m). But by using the novel property of the cosmic string, the bending angle of the Airy beam can be enlarged without any distortion of the field pattern. Figure 5j shows an Airy beam is bent with $\theta_4 = 11.5°$ by a cosmic string with mass density $\eta = 1.068$. The simulation in Fig. 5n also shows the same bending angle.

**Discussion**

In the above results, we have experimentally demonstrated the optical analogy of topological space of cosmic string by constructing an artificial waveguide based on rotational metasurface. We have observed robust definite photon deflection in both positive and negative topological space. By tuning the waveguide thickness and material loss, we can obtain the symmetry breaking of the photonic mode and mimic the creation of a cosmic string in the early universe. This remarkably robust photon deflection can be used as a new kind of omnidirectional lens, which can bend light definitely without changing the beam profiles. Such a unique property can be used in optical focusing, imaging, and information transfer. In this work, we only mimic the 1-dimensional topological defect, a cosmic string. In future, the reported method can possibly be extended to mimic a 0-dimensional monopole and 2-dimensional topological domain wall[10]. By combining the quantum field and gravity, topological gravity plays a very important role in modern theoretical physics[9]. But most of its subtle theoretical predictions are still very obscure and far from an experimental test. The method employed in this work can provide an experimental platform to investigate topological gravity and many other unusual topological properties

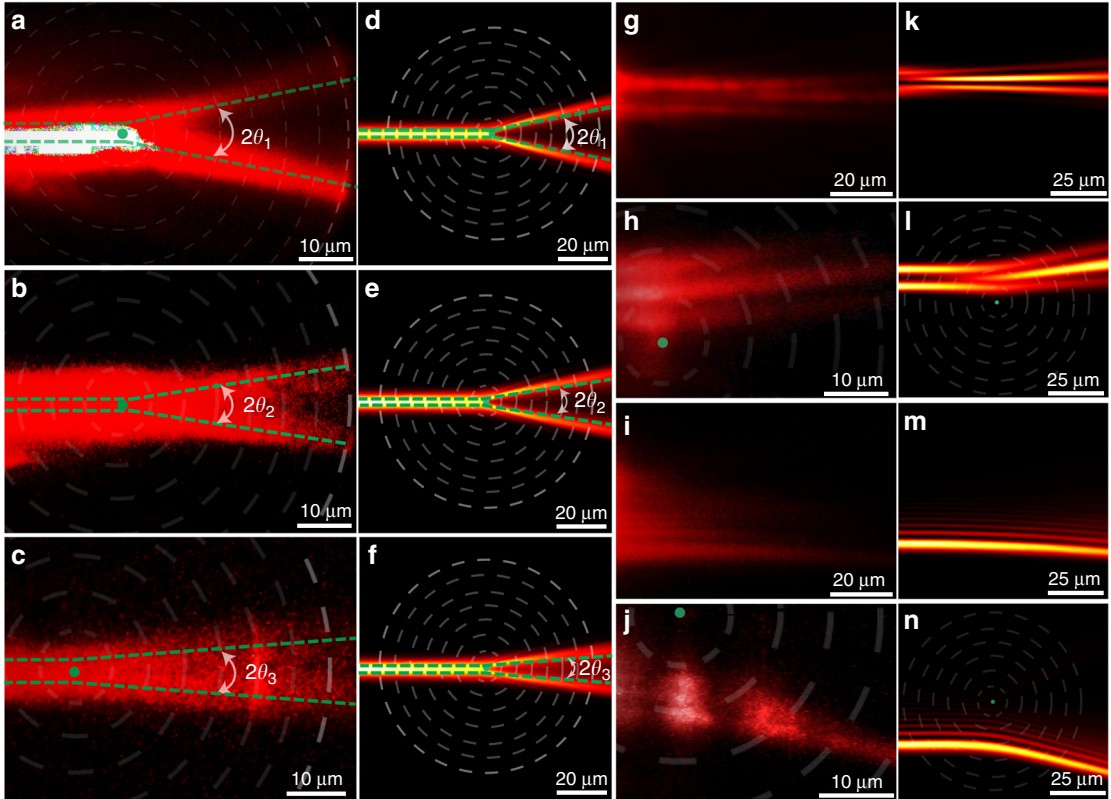

**Fig. 5** Photon deflections in different topological spaces. **a–c** Experiments of definite optical deflection by cosmic strings with different mass densities $\eta \approx$ 1.065, 1.05, 1.025, and the measured deflection angles are $\theta_1 = 11.0°$, $\theta_2 = 8.6°$, $\theta_3 = 4.4°$. For these three samples, the period of the circular grating $p =$ 120 nm, the silver filling ratio $f = 0.58$, and PMMA layer thickness $t = 300$ nm, but with different metasurface depth $h = 60, 52, 36$ nm. **d–f** The corresponding calculated results with COMSOL full-wave simulation. Experiment results of optical interference in non-trivial flat space (**g**) and topological space of cosmic string (**h**). Experiment results of the acceleration beam in flat space (**i**) and topological space of cosmic string (**j**). **k–n** Corresponding simulation results using COMSOL software. In order to magnify the detail of the optical field profile and to easily determine the deflection angle, the size of experiment images is zoomed in and the brightness contrast is increased

of spacetimes. Furthermore, the topological properties of space-time originate from the quantum effect of gravity. In order to explore the quantum effect of gravity, we should combine quantum optics with transformation optics in future research. Such prospective research may have potential applications in the relativistic quantum information technologies.

## Data availability

The data that support the findings of this study are available from the corresponding author upon reasonable request.

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

## Acknowledgements

This work was financially supported by the National Natural Science Foundation of China (Nos. 11690033, 61425018, 11621091, and 11704181), National Key R&D Program of China (2017YFA0303702), and National Key Research and Development Program of China (No. 2017YFA0205700). C.S. gratefully acknowledges the support of the National Postdoctoral Program for Innovative Talents (BX201600070).

## Author contributions

C.S., H.L., and S.N.Z. proposed and carried out the experiment. C.S., H.L., and H.Y.C. contributed to the experimental characterization and interpretation, and proposed and developed the theory. C.S., H.L., H.Y.C., and S.N.Z. co-wrote the manuscript.

## Additional information

**Competing interests:** The authors declare no competing interests.

