## [Peer Review File · Nature Communications]

Reviewers' comments:

Reviewer #1 (Remarks to the Author):

This paper reports on an interesting optical experiment, which mimics some features of hypothetical cosmic strings. The paper is well written and gives considerable background information on the recent progress in metamaterial optics related to analog spacetime models. Before publication the Authors must address the issue of reduced material parameters implemented in their experimental model. They correctly state that in order to maintain analogy with cosmic strings it is necessary for the corresponding tensor components of ϵ and μ to be equal to each other. In the discussion following eq.(2) without much explanation or justification they switch to a different situation. They assume that they can represent a cosmic string using an artificial medium having $\mu=1$, while making only the components of ϵ tensor coordinate-dependent. This is a drastic step, which needs considerable justification in the revised manuscript. If the Authors will make such a revision, this manuscript may be eventually published in Nature Communications.

Reviewer #2 (Remarks to the Author):

Current research in metamaterials has spawned significant interest in designing optical analogue gravity systems such as presented in this paper. A spacetime metric of interest can be mapped to optical medium constitutive parameters that can be accessed and mimicked by the flexibility offered by meta-media. The current manuscript does this by relating the metric describing the space around a cosmic string to a metasurface-engineered inhomogeneous, anisotropic planar waveguide, and comparing observed ray deflections with simulations. There is an interesting discussion in which genesis of a cosmic string is related to the experimental realisation in which the propagation distance in the presence of loss is identified as a symmetry-breaking parameter with respect to the coherence length of the supported guided wave modes. The paper is in my view of potential interest to a fairly broad audience.

However, I have two significant concerns:

- a) The paper consistently confuses and blurs the distinction between Analogue Gravity and Transformation Optics. Analogue gravity maps the metric of spacetime to electromagnetic medium constitutive parameters. Transformation optics takes a deformation of space and by comparing active and passive interpretations, prescribes a medium in which the transformation (such as the famous cloaking transformation) can be realised. By construction, transformation optics cannot induce curvature. Rather, it takes the initial space and represents it in a different, and potentially interesting way. The curves defining an electromagnetic cloak, regarded as geodesics, are just as 'straight' and 'parallel' as the untransformed linear parallel rays. The distinction between analogue gravity and transformation optics is especially important in the current manuscript, because of the nature of what is being modelled, i.e. the spacetime around a cosmic

$$ds^2 = dt^2 - dr^2 - \alpha^2 r^2 d\phi^2 - dz^2 \quad (1)$$

This, as the manuscript correctly notes, is just the Minkowski metric with a scaled (i.e. transformed) angular coordinate. The effect of scaling ϕ is to change the spatial topology from being planar to being conical (again as noted by the authors). The analogue that is being modelled is therefore that of gravity-induced change in the spatial topology, while the space itself (except for the singularity at the origin) remains flat. There are many instances in the manuscript that contradict this, e.g. line 17: 'The photon deflection in the topological curved space...'. All instances should be appropriately clarified before the paper can be considered to be set in its proper context. The invariance of the optical deviation with impact parameter becomes natural enough when the rays are regarded as propagating on an effective conical surface.

- b) My second concern is in the quality of the experimental results. The calculations of course show cleanly the opposite deflections for positive and negative defects, but the experimental results are not decisively convincing. I think this is true of both figures presented (Figs. 3 and 5), but comparing, say, the experimental results of Figs. 5(a)-(d) with the theoretical calculations of Figs. 5(e)-(h) we have:

The supporting text (line 281) says ‘Clearly the theoretical and experiment results agree well’. I’m not sure many readers would be convinced.

The key question for me is how will the manuscript look once a) has been addressed? It’s difficult, as I do feel that this is interesting work. On balance, though, I’m inclined to recommend against publication at this stage, perhaps with encouragement for the authors to resubmit once *both* a) and b) have been addressed.

Response Letter

Response to Reviewer #1:

Reviewer #1: "This paper reports on an interesting optical experiment, which mimics some features of hypothetical cosmic strings. The paper is well written and gives considerable background information on the recent progress in metamaterial optics related to analog spacetime models."

Our response:

We thank the referee for this positive comment on our work. The analog spacetime in metamaterials is quite fantastic topic, which has research value in both fundamental sciences and practical applications. In recent years, there are some new progress on this topic. All these works are very interesting and make some contribution to this topic. In our paper, we try to provide information to help the readers learning the background of this field. We hope this can arouse some more research interest in this field.

Reviewer #1: "Before publication the Authors must address the issue of reduced material parameters implemented in their experimental model. They correctly state that in order to maintain analogy with cosmic strings it is necessary for the corresponding tensor components of ϵ and μ to be equal to each other. In the discussion following eq.(2) without much explanation or justification they switch to a different situation. They assume that they can represent a cosmic string using an artificial medium having $\mu=1$, while making only the components of ϵ tensor coordinate-dependent. This is a drastic step, which needs considerable justification in the revised manuscript."

Our response:

We thank the referee for this good suggestion. We are sorry to forget providing enough information to justify the reduced materials parameters implemented in our experimental model. In our experiment, the different polarization states play a very important role to obtain positive and

negative topological space. When we employ equation (2) to design practical structures, we can consider the reduced effective parameters for two different polarized waves separately. In our theoretical model, two orthogonal polarization waves were defined in the cylindrical coordinate system as transverse electric (TE) wave with field (E_ϕ, E_r, H_z) and transverse magnetic (TM) wave with field (H_ϕ, H_r, E_z) . For TE wave, we can take the refractive index as $(n_\phi^2 = \epsilon_r \mu_z = \alpha^2, n_r^2 = \epsilon_\phi \mu_z = 1)$. For TM wave, we can take the refractive index as $(n_\phi^2 = \mu_r \epsilon_z = \alpha^2, n_r^2 = \mu_\phi \epsilon_z = 1)$. Thus, for both two polarized waves, the corresponding reduced refractive indexes for the cosmic string are equivalent to a uniaxial crystal with a refractive index $n = \begin{pmatrix} n_\phi & 0 \\ 0 & n_r \end{pmatrix}$. Here, we only need to control the effective refractive index tensor n to emulate cosmic string, the original required condition $\mu=1$ is not necessary and can be ignored. In practical experimental designing, we use metasurface to control the refractive index of TE and TM waveguide modes to obtain the anisotropic effective index required by cosmic string, which is presented in supplementary part II.

Reviewer #1: "If the Authors will make such a revision, this manuscript may be eventually published in Nature Communications."

Our response:

As the referee required, we have added the information about the effective index for two different polarized waves. For both TE and TM waves, the reduced material parameters can be used to emulate cosmic string. In the experiments, we only need to control the refractive index of TE and TM waveguide modes with the help of metasurface. The description is added to main text and supplementary part I and II. We hope these added information can make our theoretical model more understandable and meet the referee's requirement.

Response to Reviewer #2:

Reviewer #2: “Current research in metamaterials has spawned significant interest in designing optical analogue gravity systems such as presented in this paper. A spacetime metric of interest can be mapped to optical medium constitutive parameters that can be accessed and mimicked by the flexibility offered by meta-media. The current manuscript does this by relating the metric describing the space around a cosmic string to a metasurface-engineered inhomogeneous, anisotropic planar waveguide, and comparing observed ray deflections with simulations. There is an interesting discussion in which genesis of a cosmic string is related to the experimental realisation in which the propagation distance in the presence of loss is identified as a symmetry-breaking parameter with respect to the coherence length of the supported guided wave modes. The paper is in my view of potential interest to a fairly broad audience. “

Our response:

Thanks for the referee’s positive comments on our work. We totally agree with the referee that current research in metamaterials has spawned significant interest in designing optical analogue gravity systems. In our knowledge, theorists have proposed many interesting models to mimic gravity in metamaterials. But there is a big challenge to employ these models in practical systems due to the limitation of experiment techniques. For an instance, although some theory papers have proposed interesting methods to mimic cosmic string in metamaterials, there is no experimental work to realize cosmic string up to now. On the other hand, 2D metasurfaces have become a very hot topic in optical community due to its smaller loss and easier fabrication compared with 3D bulk metamaterials. Metasurfaces provide us a new possible technique to mimic gravity in optics. With the help of metasurface, our work reports an artificial waveguide to mimic cosmic string. The definite light deflection induced by topological defects was firstly observed directly in the experiments. Generally, we are sure such metasurface-engineered waveguides provide a flexible experimental platform to mimic many other gravitational effects in optics. This work will make a contribution to the progress of experiments in optical analogue gravity.

Reviewer #2: “The paper consistently confuses and blurs the distinction between Analogue Gravity and Transformation Optics. Analogue gravity maps the metric of spacetime to electromagnetic medium constitutive parameters. Transformation optics takes a deformation of space and by comparing active and passive interpretations, prescribes a medium in which the transformation (such as the famous cloaking transformation) can be realised. By construction, transformation optics cannot induce curvature. Rather, it takes the initial space and represents it in a different, and potentially interesting way. The curves defining an electromagnetic cloak, regarded as geodesics, are just as ‘straight’ and ‘parallel’ as the untransformed linear parallel rays. The distinction between analogue gravity and transformation optics is especially important in the current manuscript, because of the nature of what is being modelled, i.e. the spacetime around a cosmic string. The metric used (Eq. (1)) is a limiting form first derived by Vilenkin [Phys. Rev. D 23, 852 (1981)] in the low-mass limit, $G\mu \ll 1$, i.e.

$$ds^2 = dt^2 - dr^2 - \alpha^2 r^2 d\phi^2 - dz^2 \quad (1)$$

This, as the manuscript correctly notes, is just the Minkowski metric with a scaled (i.e. transformed) angular coordinate. The effect of scaling ϕ is to change the spatial topology from being planar to being conical (again as noted by the authors). The analogue that is being modelled is therefore that of gravity-induced change in the spatial topology, while the space itself (except for the singularity at the origin) remains flat. There are many instances in the manuscript that contradict this, e.g. line 17: ‘The photon deflection in the topological curved space....’. All instances should be appropriately clarified before the paper can be considered to be set in its proper context. The invariance of the optical deviation with impact parameter becomes natural enough when the rays are regarded as propagating on an effective conical surface.”

Our response:

We thank the referee for the instructive comments. We admit that there is a distinction between Analogue Gravity and Transformation Optics. In Einstein’s general relativity theory, the local spacetime curvature is

determined by the local energy and momentum. Strictly, curved space in gravity should be the space with local curvature. In transformation optics, the local curvature is not required by most of applications, such as invisible cloak. Therefore, the transformation optical medium without local curvature cannot be regarded as analogue of curved space. In gravity, cosmic string is a very special case which only possess local curvature at the origin singularity. The space around the string has no local curvature. Strictly speaking, the space around the cosmic string is flat space except the origin singularity. The photon deflection induced by cosmic string is not caused by local curvature, but by the global topological property. Therefore, the description “photon deflection in curved space” is not accurate. In order to avoid this confusion, the word “curved space” improperly used in our paper has been replaced by some other proper expressions. The distinction between Analogue Gravity and Transformation Optics was carefully taken care in the revised manuscript. We also thank the referee for the suggested reference about metric of cosmic string in the low-mass limit. We add it in the text as Ref. 82.

Reviewer #2: “My second concern is in the quality of the experimental results. The calculations of course show cleanly the opposite deflections for positive and negative defects, but the experimental results are not decisively convincing. I think this is true of both figures presented (Figs. 3 and 5), but comparing, say, the experimental results of Figs. 5(a)-(d) with the theoretical calculations of Figs 5(e)-(h) we have... The supporting text (line 281) says ‘Clearly the theoretical and experiment results agree well’. I’m not sure many readers would be convinced.”

Our response:

We admit that some figure given in origin manuscript are not quite clear to show convinced information. The reasons reducing the quality of experiment results are mainly caused by the experiment techniques used in our work. Firstly, in order to observe the deflection of laser beam inside the polymer waveguide, we doped many quantum dots inside the PMMA layer. When a light is coupled into the waveguide, it will excite the quantum dots and emit visible fluorescence. Through imaging with fluorescent signal, we

can observe the photon deflection around the cosmic string. During the quantum dots doping process, the quantum dots are not distributed uniformly inside the PMMA. Some quantum dots will aggregate together and make the fluorescent intensity emitted at different locations non-uniform. This is one reason that reduce the quality of the imaging. Secondly, we employ focused ion beam (FIB) to fabricate the metasurface in experiment. In the designed structures, the period of metasurface is only 160nm and the slit width is only 70nm. The size of the structure is very small and close to the ultimate machining precision of FIB. Some structure defects were unavoidably induced in the fabrication process. These defects cause random scattering of light inside the waveguide. This is another reason reduced the quality of imaging. Although the quality of experiment picture is not ideal due to the above experiment problems, the obtained images can still provide us enough information to confirm our theoretical results. In order to make the imaging detail clearer, we change the size scale of the pictures and magnify the details of the deflected laser beams in the images. In the new revised figures (such as figure 3 and 5 in the new manuscript), the quality is improved and the photon deflection angles can be well determined. In order to help readers to understand the experimental results, we also add some text to explain the quality of experiment images. We hope the improved experiment figures and added explanations can meet the referee's requirement and provide enough information to support our conclusions. Furthermore, we are sure it is possible to get better quality pictures after the experiment techniques are improved in the future research.

Reviewer #2: "The key question for me is how will the manuscript look once a) has been addressed? It's difficult, as I do feel that this is interesting work. On balance, though, I'm inclined to recommend against publication at this stage, perhaps with encouragement for the authors to resubmit once both a) and b) have been addressed."

Our response:

We thank again for the referee's comments on our work and suggestions to improve the quality of our paper. We consider the questions seriously and

make the responses carefully. Some corresponding changes are made in the manuscript. In response to question (a), some improper usages of “curved space” are replaced by some other proper expressions. In response to question (b), the size scale of some figures is magnified to make the experimental detail clearer. Some explanations are added to help readers to understand the experimental results. We hope our responses and changes are able to satisfy the referee’s requirement and new revised manuscript can be accepted by Nature Communications.

Change List

- (1) To address the reviewer #1 comments, we have added a new sentence on page 4 of the revised manuscript which states: “Here, considering different polarized light, we can define two orthogonal polarized waves: transverse electric (TE) wave with field (E_ϕ, E_r, H_z) and transverse magnetic (TM) wave with field (H_ϕ, H_r, E_z) . For TE wave, we can take the refractive index $(n_\phi^2 = \epsilon_r \mu_z = \alpha^2, n_r^2 = \epsilon_\phi \mu_z = 1)$. For TM wave, we can take the refractive index $(n_\phi^2 = \mu_r \epsilon_z = \alpha^2, n_r^2 = \mu_\phi \epsilon_z = 1)$. Thus, for both two polarizations, the corresponding refractive indexes of the cosmic string are equivalent to a uniaxial crystal with a rotating axis $n = \begin{pmatrix} n_\phi & 0 \\ 0 & n_r \end{pmatrix}$.” The sentence clarifies the reduced material parameters which are used to emulate cosmic string.
- (2) To address the reviewer #1 comments, we have updated our reference list to include one paper about reduced material parameters (J. Opt. 20, 2040 (2018)), which is cited as ref. [85].
- (3) To address the reviewer #2 comments, the title of our paper has been changed to “Definite Photon Deflections in Topological Metasurface-engineered Waveguides and Symmetry Breaking Phase Transitions with Material Loss”.
- (4) In response to reviewer #2, the improper usages of “curved space” have been replaced by other proper expressions in the revised manuscript.
- (5) In response to reviewer #2, we have updated our reference list to include one paper about metric of cosmic string in the low-mass limit (*Phys. Rev. D* 23, 852 (1981)), which is cited as ref. [82].
- (6) In response to reviewer #2, we have changed the size scale of the

experimental images and magnify the details of the deflected laser beams in the waveguides (see the experimental images in figure 3 and 5).

(7) In response to reviewer #2, we have added a paragraph on page 6 explaining the experimental processes which would reduce the quality of experiment results. The added paragraph reads: "Compared with calculation results, the experiment results are not very clear which is mainly caused by the experiment techniques used in our work. Firstly, the quantum dots are not distributed uniformly inside the PMMA layer. Some quantum dots aggregate together and make the fluorescent intensity emitted at different locations non-uniform. Secondly, the size of the structure is very small and close to the ultimate machining precision of FIB. Some structure defects are unavoidably induced in the fabrication process. These defects will cause random scattering of light inside the waveguide. Although the quality of experiment picture is not ideal due to the above experiment process, the deflection angles can be still determined and compared with theoretical results."

(8) In response to reviewer #2, we have changed the expressions about the comparison between experimental and simulation results. Sentence on page 7: "Although the experiment results are not very clear due to experimental technique problems, it can be still seen that the deflection angles do not change under three different incident impact parameters which agree with theoretical and simulation results."; Sentence on page 9: "Although the experimental patterns are not very clear, we can still determine the deflection angles which agree with the theoretical and numerical results."; Sentence on page10: "The simulations in figure 5(n) also shows the same bending angle."

Reviewers' comments:

Reviewer #1 (Remarks to the Author):

I agree with Referee #2 that agreement between experiment and theory remains to be a concern. For example, I do not see a good match between the following pairs of experimental and theoretical images:

Fig.3g and Fig.3j, Fig.3h and Fig.3k, Fig.3i and Fig.3l, Fig.5c and Fig.5f, Fig.5h and Fig.5l, Fig.5j and Fig.5n. Can the Authors improve their experimental images before publication? Otherwise, the paper will remain to be unconvincing

Reviewer #2 (Remarks to the Author):

In their response the authors acknowledge the distinction between transformation optics and analogue gravity. Mostly the revised manuscript has replaced the phrases that blurred this distinction, with ones that respect it. However, I am still concerned with the passage in the second paragraph:

Fortunately, analog models from various systems in the laboratory environment have been motivated by the possibility of investigating phenomena not readily accessible in their cosmological counterparts. Examples are Hawking-Unruh radiation emulated from a Fermi-degenerate liquid¹⁴, a superconductor¹⁵, optical fiber¹⁶, nonlinear crystal¹⁷, ion ring¹⁸ and Bose-Einstein condensates¹⁹. Recently the ongoing revolution in transformation optics based on metamaterials²⁰⁻³⁸ that manipulate permittivity and permeability profiles can be extensively investigated to design artificial materials to mimic general relativity phenomena. Examples from General Relativity are black holes³⁹⁻⁴⁶, Einstein ring⁴⁷, cosmic string⁴⁸⁻⁵⁰, Minkowski spacetimes⁵¹, electromagnetic wormholes⁵², De-Sitter Space^{53,54}, cosmological inflation, and redshift^{55,56}. The underlying principle is the form invariance of Maxwell's equations between the complex inhomogeneous media and the background of an arbitrary spacetime metric.

As it stands, the highlighted sentence and those preceding and following clearly imply a correspondence between analogue gravity and transformation optics. I feel this should be addressed before the paper can be accepted. It might be a good opportunity for the authors to explain and clarify the distinction in the text, as this has often been misunderstood in other literature.

My second concern was with the quality of the experimental results. The results in the revised manuscript are clearer, and the authors have provided explanations in the text as to the experimental challenges to produce clean data. I am not an experimentalist, but I can well believe that the presented results are at the limit of technological challenge, and I am prepared to accept that without the issue further holding up publication.

Response Letter

Response to Reviewer #1:

Reviewer #1: “I agree with Referee #2 that agreement between experiment and theory remains to be a concern. For example, I do not see a good match between the following pairs of experimental and theoretical images: Fig.3g and Fig.3j, Fig.3h and Fig.3k, Fig.3i and Fig.3l, Fig.5c and Fig.5f, Fig.5h and Fig.5l, Fig.5j and Fig.5n. Can the Authors improve their experimental images before publication? Otherwise, the paper will remain to be unconvincing.”

Our response:

We admit that, compared with other experiment images, figure 3 (g), (h), (i) and figure 5 (c), (h), (j) are not so so perfect to match theoretical images. In theoretical images, the field profiles of optical beams are smooth and clear. While, the optical field profiles in these experimental images are rough and the details of optical beam are not clear. These quality problems are caused by the limitation of experimental techniques used in our sample fabrications and optical measurements.

As referee required, we have tried to improve the experimental images as clear as possible (see the revised figure 3 (g), (h), (i) and figure 5 (c), (h), (j)). In order to make these images clearer, the size of experiment images is zoomed in and the brightness contrast of the images is increased. In this way, we magnify the detail of optical field profile and make optical deflection beam more visible.

In this work, the most remarkable optical effects in topological space of cosmic string is the definite photon deflection angle which is robust and independent on the input location of optical beam. Therefore, the most important thing is to compare the deflection angles in the theoretical and experimental images. Although the improved experiment images are still not so perfect, the deflection angles of optical beam, which are marked with green dashed lines in the figures, can be clearly determined and compared with theoretical results. The results show that the measured deflection angles in all these experimental images match the corresponding angles in theoretical images. We hope the improved figure 3 and 5 can provide convincing information and meet the referee's requirement.

Response to Reviewer #2:

Reviewer #2: “In their response the authors acknowledge the distinction between transformation optics and analogue gravity. Mostly the revised manuscript has replaced the phrases that blurred this distinction, with ones that respect it. However, I am still concerned with the passage in the second paragraph:

Fortunately, analog models from various systems in the laboratory environment have been motivated by the possibility of investigating phenomena not readily accessible in their cosmological counterparts. Examples are Hawking-Unruh radiation emulated from a Fermi-degenerate liquid¹⁴, a superconductor¹⁵, optical fiber¹⁶, nonlinear crystal¹⁷, ion ring¹⁸ and Bose-Einstein condensates¹⁹. ***Recently the ongoing revolution in transformation optics based on metamaterials²⁰⁻³⁸ that manipulate permittivity and permeability profiles can be extensively investigated to design artificial materials to mimic general relativity phenomena.*** Examples from General Relativity are black holes³⁹⁻⁴⁶, Einstein ring⁴⁷, cosmic string⁴⁸⁻⁵⁰, Minkowski spacetimes⁵¹, electromagnetic wormholes⁵², De-Sitter Space^{53,54}, cosmological inflation, and redshift^{55,56}. The underlying principle is the form invariance of Maxwell’s equations between the complex inhomogeneous media and the background of an arbitrary spacetime metric.

As it stands, the highlighted sentence and those preceding and following clearly imply a correspondence between analogue gravity and transformation optics. I feel this should be addressed before the paper can be accepted. It might be a good opportunity for the authors to explain and clarify the distinction in the text, as this has often been misunderstood in other literature.”

Our response:

As the referee required, the introduction part is rewritten to emphasize the distinction between analogue gravity and transformation optics. Especially, the highlighted sentence is changed. The implication of direct correspondence between analogue gravity and transformation optics is carefully avoided in the text. In order to explain and clarify the distinction between analogue gravity and transformation optics, the local curvature required by analogue gravity is especially addressed in the text, which is often neglected by other literatures. This will help the readers realize the space curvature is the key characteristics in general relativity and analogue gravity.

Reviewer #2: “My second concern was with the quality of the experimental results. The results in the revised manuscript are clearer, and the authors have provided explanations in the text as to the experimental challenges to produce clean data. I am not an experimentalist, but I can well believe that the presented results are at the limit of technological challenge, and I am prepared to accept that without the issue further holding up publication.”

Our response:

We are appreciated that the referee understands the temporary limitation of experiment techniques used in our work and agree the improved experiment images are clearer than original ones. At present, the employment of metasurfaces to transformation optics is a quite new technique, especially in visible frequency range. There are still many challenges in materials fabrications and optical measurements. We hope some new methods can be found to improve experiment techniques and obtain better experiment results in the future.

Changes Lists

- (1) The second paragraph is changed. This passage

“Fortunately, analog models from various systems in the laboratory environment have been motivated by the possibility of investigating phenomena not readily accessible in their cosmological counterparts. Examples are Hawking-Unruh radiation emulated from a Fermi-degenerate liquid¹⁴, a superconductor¹⁵, optical fiber¹⁶, nonlinear crystal¹⁷, ion ring¹⁸ and Bose-Einstein condensates¹⁹.”

is moved from second paragraph to the first paragraph.

- (2) In second paragraph, the highlighted sentence by referee #2

“Recently the ongoing revolution in transformation optics based on metamaterials²⁰⁻³⁸ that manipulate permittivity and permeability profiles can be extensively investigated to design artificial materials to mimic general relativity phenomena.”

is changed to

“On the other hand, transformation optics based on metamaterials that manipulate permittivity and permeability profiles can be extensively investigated to design many artificial materials with novel optical applications²⁰⁻³⁸. The most important application of transformation optics is invisible cloaks, in which light is regarded as linear parallel geodesic rays in deformed spaces. In Einstein’s general relativity theory, the spacetime curvature is determined by the energy and momentum. Recently, through mapping the metric of spacetime to electromagnetic medium constitutive parameters with local curvature, the ongoing revolution in metamaterials can be extensively designed to mimic general relativity phenomena.”

- (3) In figure 3, the experiment images (g) (h) (i) is improved
(4) In figure 5, the experiment images (c) (h) (j) is improved.